⊘ | **Open Peer Review** | Ecology | Research Article

# Compared with pure forest, mixed forest alters microbial diversity and increases the complexity of interdomain networks in arid areas

Qian Guo,[1,2,3] Lu Gong[1,2,3]

**ABSTRACT** Soil microbial communities are key to material cycling and energy exchange in forest soils. As the Tianshan Mountains in Xinjiang are rich in tree species, it is important to understand the impact of tree species composition on soil microbial community structure and co-occurrence networks in arid mountain habitats. In this study, soil microbial community structure of five typical vegetation types in the Tianshan Mountains of Xinjiang, China, was investigated using bacterial 16S rRNA and fungal ITS sequences. The results showed that the relationship between the fungal community and tree species was strong, while the bacterial community was mainly affected by soil carbon. The dominant bacterial composition was Ascomycota, which had the highest relative abundance in herbaceous soils (27%–41%); the dominant fungal phylum was Ascomycota (50.22%–63.95% of the total sequences), followed by Stramonium (8.99%–28.09%). The fungal community structure of mixed coniferous and broadleaf forests was similar to that of conifers, while the bacterial community structure was similar to that of pure broadleaf forests. Mixed forests (mean connectivity: 812.50, clustering coefficient: 8.89) showed a more complex co-occurrence network and a significant increase in microbial structure related to soil carbon and nitrogen than pure forests (broad: mean connectivity: 710.69, clustering coefficient: 6.94; coniferous: mean connectivity: 730.79, clustering coefficient: 7.76). The results contribute to the understanding of the interactions between soil microorganisms and soil factors in different forest ecosystems and provide new insights into the interactions among forest microorganisms in arid zones.

**IMPORTANCE** The results provide a comparative study of the response of soil microbial ecology to the afforestation of different tree species and deepen the understanding of the factors controlling soil microbial community structure.

**KEYWORDS** interdomain networks, material cycling, soil microbial community, Tianshan Mountains, vegetation type

Address correspondence to Lu Gong, gonglu721@163.com.

The authors declare no conflict of interest.

See the funding table on p. 17.

Soil microorganisms are important components of forest ecosystems, and different tree species (1) have different functional characteristics (such as phylum and leaf traits) that can modify the structure of soil microbial communities and general patterns of soil nutrient cycling. Previous studies have explored the important role of ground vegetation and soil properties (2) in driving microbial communities and soil characteristics, especially pH, which is often reported as a powerful driving factor for microbial diversity (3). However, studies have shown that tree species have a greater impact on community structure than the soil environment at a small scale (1, 4). Bacteria and fungi exhibit different response patterns, depending on various factors. Naz et al. (5) noted that bacteria are mainly influenced by soil environmental factors, especially pH (6), whereas fungi, owing to their ability to form mycorrhizal roots within the root system, are

largely influenced by the dominant tree species (7). Understanding the composition and interaction of fungi and bacteria under different tree species, and exploring the factors controlling the microbial community structure can reveal key ecosystem processes in forest cycles, thereby providing a basis for forest management. However, the mechanisms by which different tree species and their compositions affect soil community processes are currently poorly understood (8). Due to the large amount of microbial data and the complexity of internal processes in soils, network analysis is gradually being applied in order to further quantify the controlling factors of microbial processes. Compared to previous studies of soil microorganisms, which usually only constructed a single network of bacteria or fungi, this study incorporated bacteria and fungi into the microbial-soil interdomain network, which is closer to the real biological-environmental interaction and advances the understanding of the microbial interaction network in arid areas.

Under similar climatic and local site conditions, different tree species exhibit variations in traits such as the quantity and chemical composition of root exudates, leaf morphology and characteristics (9), and the ability to redistribute nutrients (10), which directly affect the number and distribution of soil microorganisms by affecting soil structure, physicochemical properties, and enzyme activity, altering the interboundary signal exchange between plants and microorganisms and redistributing nutrients. Previous studies have shown that broad-leaved forests tend to have a higher abundance of Gram-negative bacteria, while coniferous forests harbor more fungi and actinomycetes (11). In contrast, mixed forests display diversified soil resource pools, high soil heterogeneity, reduced competition pressure within the microbial community, and high diversity.

Soil microbes are the most abundant component of terrestrial biodiversity and play an important role in ecosystem processes such as carbon and nitrogen cycling and soil formation. According to the habitat division hypothesis, the microhabitats created by different vegetation types select for different microbial colonization, which play corresponding roles in the ecosystem through different ecological strategies. Tree species assemblages help to form dense roots and increase nutrient release, thereby increasing soil fungal activity, further complicating fungal community structure and enhancing their ability to adapt to environmental changes. Fungi produce a series of hydrolytic and oxidative enzymes that break down dead or senescent plant residues and increase soil organic matter, accelerating organic matter decomposition and nutrient cycling. In contrast, bacteria are indispensable for the decomposition of easily degradable organic matter and rapid carbon cycling media. The compositional structure of microorganisms reflects changes in nutrient availability and interspecies competition, and altered bacterial:fungal ratios cause different contributions of microbial catabolism and anabolism to soil carbon and nitrogen dynamics, which may lead to asynchronization of soil carbon and nitrogen (12).

Vegetation types indirectly affect microorganisms through their influence on soil properties. Owing to the disparities between soil bacterial and fungal communities (such as morphology and resource requirements), their response patterns to soil factors differ (13). Bacteria have a low carbon (C)-nitrogen (N) biomass and a narrow pH tolerance range. In contrast, fungi have a higher N demand, are reliant on plant products or exudates, are more sensitive to changes in soil nutrients, and have stronger aboveground interactions (14). Soil C, N, and phosphorus (P) is an essential element in the growth of microorganisms, and soil organic matter content strongly impacts microbial abundance and the fungal:bacterial ratio (15). A previous study found that when soil N is sufficient, plants have less dependence on mycorrhizal fungi and reduce the amount of carbon flowing into the rhizosphere (16), leading to a decrease in mycorrhizal fungi. Microbial processes in arid areas are sensitive to fluctuations in water content, which affects nutrient fluidity and plant input. Soil moisture levels can affect the alpha diversity of microbial communities (17). It is important to note that when soil moisture decreases,

even if the root secretion rate does not decrease, the absorption of root exudate C by bacteria also decreases (18).

The Tianshan Mountains (40°N~44°N, 70°E~90°E) are the largest mountain system in Central Asia, and the Tianshan forests play a leading role in water conservation, soil and water conservation, and the formation and maintenance of other ecosystems. The composition of different tree species within these forests is expected to alter soil characteristics and further affect soil microbial communities. Therefore, this study focused on different vegetation types (i.e., coniferous forest, broad-leaved forest, mixed coniferous and broad-leaved forest, shrubbery, and herbs as the control) as research objects. The study aimed to investigate the impact of these vegetation types on the composition and network dynamics of soil microbial communities in typical Tianshan Mountain forests. These findings will provide a scientific basis for understanding the microbial structure and function of mountain forest ecosystems in arid areas, as well as their interaction mechanisms. Based on the above analysis, he following scientific questions are proposed: (i) does the composition of different tree species in arid mountainous forests affect the structure of microbial communities? (ii) are there differences in soil microbial co-occurrence networks under different vegetation types? (iii) is there a correlation between microbial community composition and soil factors?

## RESULTS

### Physicochemical properties of forest soil of different tree species

Significant differences in soil physicochemical characteristics among different vegetation types are shown in Table 1. In the pure forest, the soil pH value (6.26–6.45) of the coniferous sample plot was significantly lower than that of the broad-leaved sample plot (6.66–6.87) ($P = 0.05$), and the pH value (6.43–6.7) of the mixed forest containing coniferous leaves was also lower than that of the elm forest. Compared to the pure coniferous forest and broad-leaved forest, the total soil carbon ($P = 0.05$) in the mixed forest increased by 12.45% and 4.14%, whereas total nitrogen (TN) ($P = 0.05$) increased by 12.1% and 22.15%, respectively. At each sampling site, the same trend was shown, and the soil C and N contents of Western Tianshan Mountains (XTS) were higher than the other two regions. Available nitrogen (AN) showed variation in only Eastern Tianshan Mountains (DTS), and the AN content in forests is obviously different from those in shrubs and herbs. Total phosphorus content was in the order of mixed forest > broad-leaved forest > coniferous forest > shrubbery > herbs, and the performance of available phosphorus was similar to that of total phosphorus. The soil moisture content of the Western Tianshan sampling site was the highest, ranging from 29.08% to 33.07%, with an average soil temperature of 13.62°C–14.4°C. Its hydrothermal conditions were optimal. In contrast, the soil moisture content of the Eastern Tianshan sampling site was the lowest, ranging from 24.51% to 26%. The electrical conductivity (EC) varied not significantly at each sampling point.

### Soil microbial diversity and composition under different vegetation types

A total of 3,781,745 high-quality bacterial sequences and 281,967 high-quality fungal sequences were obtained from 45 sequenced soil samples. A total of 75 phyla of bacteria and 19 phyla of fungi were found.

Fig. 1 shows the distribution of species with a relative abundance of bacteria and fungi exceeding 1%. There are 18 phyla of bacteria and 19 of fungi. These species accounting for approximately 85%–90% of the total number of bacteria; Proteobacteria is the main phylum of Gram-negative bacteria, accounting for 20.84%–28.98% of the total sequence in the five different vegetation types, with the highest relative abundance in herbaceous soil. Acidobacterium and actinomycetes were the main phyla of Gram-positive bacteria, accounting for 10.59%–14.66% and 9.17%–11.83% of the total sequence, respectively. The dominant fungal phylum is Ascomycota (50.22%–63.95% of the total sequence), followed by Basidiomycota (8.99%–28.09%). Species with a

**TABLE 1** Physicochemical factors of soil of different vegetation types[a]

| Site | Vegetation type | TC (mg/g) | TN (mg/g) | TP (mg/g) | PH | Wc (%) | TC (°C) | AP (mg/g) | AN (mg/g) | EC (us/cm) |
|---|---|---|---|---|---|---|---|---|---|---|
| XTS | Coniferous | 267.08 ± 11.0ab | 5.24 ± 1.06a | 1.53 ± 0.14c | 6.26 ± 0.06b | 30.43 ± 2.28b | 18.67 ± 5.42ab | 75.51 ± 19.83c | 1.05 ± 0.24a | 175.93 ± 10.28a |
|  | Broad leaf | 256.74 ± 20.58b | 6.59 ± 0.53ab | 1.73 ± 0.12b | 6.6 ± 0.05a | 34.76 ± 0.83a | 17.52 ± 1.55b | 216.50 ± 13.81b | 1.24 ± 0.17a | 193.83 ± 26.45a |
|  | Mixed forest | 297.44 ± 18.83c | 4.94 ± 0.63a | 2.17 ± 0.39a | 6.46 ± 0.14ab | 35.2 ± 2.36a | 17.79 ± 1.92ab | 254.63 ± 21.21a | 1.14 ± 0.12a | 198.53 ± 31.59a |
|  | Shrub | 161.92 ± 21.37a | 7.62 ± 2.07b | 1.72 ± 0.26b | 6.7 ± 0.09a | 22.67 ± 2.48c | 14.19 ± 2.64b | 189.52 ± 13.41b | 0.92 ± 0.08a | 186.57 ± 5.49a |
|  | Herbal | 205.04 ± 8.486d | 3.85 ± 0.18c | 1.46 ± 0.11c | 6.63 ± 0.34a | 27.64 ± 1.40b | 23.00 ± 7.13a | 66.61 ± 18.68c | 1.10 ± 0.28a | 176.50 ± 5.90a |
| DTS | Coniferous | 225.72 ± 6.05a | 4.54 ± 1.40ab | 1.63 ± 0.18a | 6.45 ± 0.07b | 25.03 ± 0.84a | 20.19 ± 3.49b | 84.55 ± 15.29b | 0.97 ± 0.05b | 223.13 ± 32.13a |
|  | Broad leaf | 241.78 ± 15.43a | 6.03 ± 1.35a | 1.72 ± 0.06a | 6.80 ± 0.03a | 32.83 ± 9.25a | 25.32 ± 1.23a | 96.24 ± 7.48a | 1.19 ± 0.16b | 193.23 ± 0.42a |
|  | Mixed forest | 238.92 ± 12.26a | 6.39 ± 0.73a | 2.19 ± 0.52a | 6.70 ± 0.15a | 29.62 ± 7.47a | 13.66 ± 0.90c | 117.50 ± 17.80a | 1.14 ± 0.17b | 207.03 ± 17.47a |
|  | Shrub | 160.60 ± 11.56b | 4.19 ± 1.03b | 1.97 ± 0.52a | 6.68 ± 0.12a | 23.13 ± 0.94a | 23.89 ± 1.58a | 78.84 ± 15.77b | 1.29 ± 0.27a | 199.93 ± 5.40a |
|  | Herbal | 164.56 ± 20.29b | 3.50 ± 1.32b | 1.60 ± 0.17a | 6.73 ± 0.12a | 23.87 ± 1.00a | 26.91 ± 0.44a | 63.69 ± 24.27b | 1.62 ± 0.25a | 210.00 ± 7.94a |
| ZTS | Coniferous | 193.16 ± 16.39ac | 6.86 ± 1.62a | 1.06 ± 0.24a | 6.42 ± 0.19b | 28.93 ± 0.15ac | 12.81 ± 0.69c | 117.37 ± 28.58a | 1.29 ± 0.23a | 251.33 ± 35.95a |
|  | Broad leaf | 249.92 ± 22.49a | 6.17 ± 1.02ab | 1.00 ± 0.03a | 6.86 ± 0.02a | 32.88 ± 3.39ac | 17.03 ± 0.85b | 120.56 ± 22.14a | 1.22 ± 0.29a | 193.93 ± 11.14a |
|  | Mixed forest | 238.92 ± 23.76ac | 7.36 ± 1.24a | 1.27 ± 0.06a | 6.42 ± 0.31b | 33.91 ± 7.59a | 18.54 ± 0.34b | 129.06 ± 16.62a | 1.49 ± 0.31a | 267.83 ± 22.32a |
|  | Shrub | 150.04 ± 10.75b | 4.32 ± 0.54b | 1.09 ± 0.63a | 6.33 ± 0.32c | 25.45 ± 1.21b | 16.80 ± 2.26b | 84.68 ± 15.77b | 1.67 ± 0.29a | 213.00 ± 11.03a |
|  | Herbal | 179.3 ± 27.08b | 3.49 ± 0.19c | 9.89 ± 0.24a | 6.31 ± 0.14c | 22.39 ± 1.38b | 25.09 ± 0.91a | 71.53 ± 17.40b | 1.32 ± 0.42a | 196.73 ± 13.76a |

[a]AN, available nitrogen; AP, available phosphorus; EC, electrical conductivity; ST, soil temperature; TC, total carbon; TN, total nitrogen; TP, total phosphorus; Wc, water content.

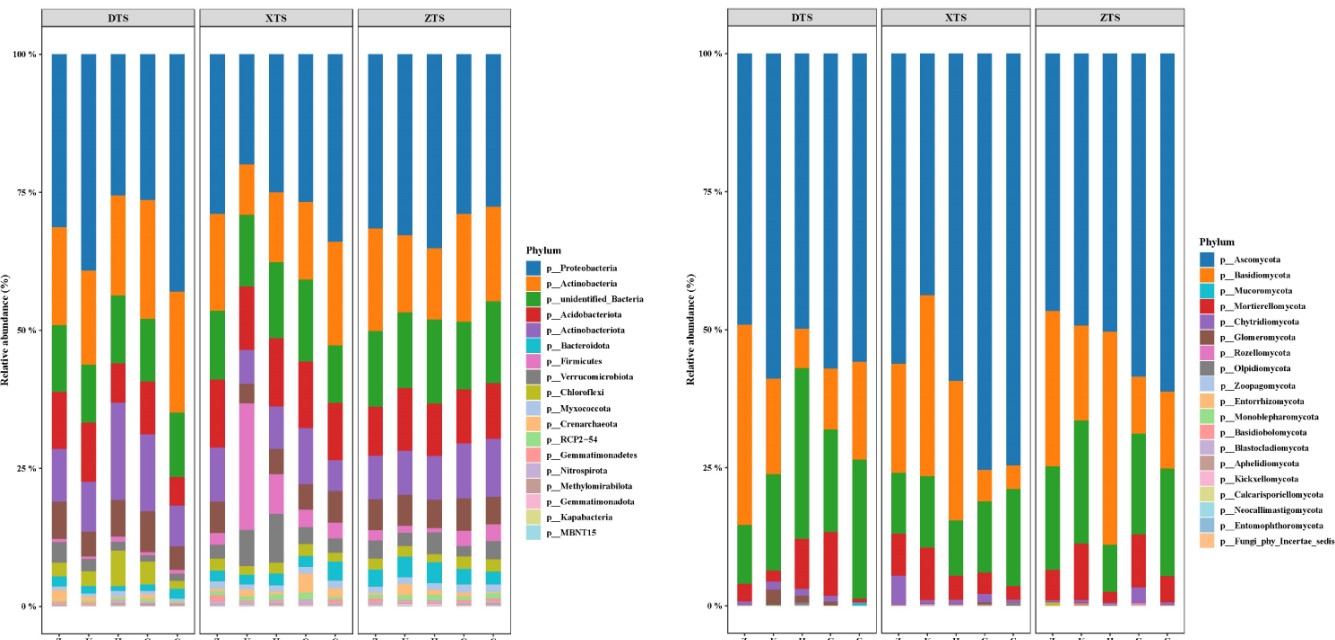

**FIG 1** Relative abundance of bacteria and fungi in the different vegetation types. C, herbal; DTS, Eastern Tianshan Mountains; G, shrub forest; H, mixed coniferous and broad-leaved forest; K, *Ulmus pumila* forest; XTS, Western Tianshan Mountains; Z, *Picea schrenkiana* forest; ZTS, Central Tianshan Mountains.

relative abundance exceeding 1% accounted for 69.0%–91.4% of the fungal communities. The abundance of Coccidiophyta, Chytridiomycota, Rozellomycota, and Mucoromycota increased significantly in broad-leaved soils. Basidiomycota and Chytridiomycota had high contents in the soil of coniferous forests, Ascomycota and Mortierellomycota accounted for the largest proportion in the shrubbery, and the microbial composition in the mixed forest was between that of broadleaf and coniferous forests.

In terms of vegetation type, there was a more significant difference between herbaceous soil and other forest soil types, with a decrease in actinomycetes and a significant increase in oligotrophic microbial communities. Actinobacteria is a copiotrophic phylum that is more abundant in broad-leaved forests with better soil nutrients. Compared with bacteria, the distribution of fungi in different types of soils is more uneven, and the two dominant fungi are more important in herbs and shrubs, crowding out the ecological niche of other fungi. The results of microbial composition analysis showed that the fungal community structure of mixed forests was more similar to that of coniferous forests (19), whereas the bacterial community structure was more similar to that of broad-leaved plants. Compared to the central and Western Tianshan Mountains, the relative abundance of microorganisms in the Eastern Tianshan Mountains exhibited significant variation. Short rhizobia exhibited significant variations at both sampling points, whereas variations in *Mycobacterium* and *Pseudomonas* were mainly concentrated in the Eastern Tianshan Mountains.

The Shannon and Chao1 indices reflect the activity, actual abundance, and uniformity of the bacteria. There were significant differences in the Shannon diversity indices of the bacterial and fungal communities among the five vegetation types ($P < 0.05$, Fig. 2). In terms of alpha diversity, these indicators show the same trend at different points, that bacteria were mixed, broad-leaved forest > coniferous forest, and shrubbery > herbage, and there were large differences in each system. The overall performance of fungi was as follows: pure coniferous forest > broad-leaved forest > shrub, herbage > mixed forest, and the variation in each type of soil was less than that of bacteria. Bacteria and fungi showed a trend of XTS > Central Tianshan Mountains (ZTS) > DTS at different sampling points.

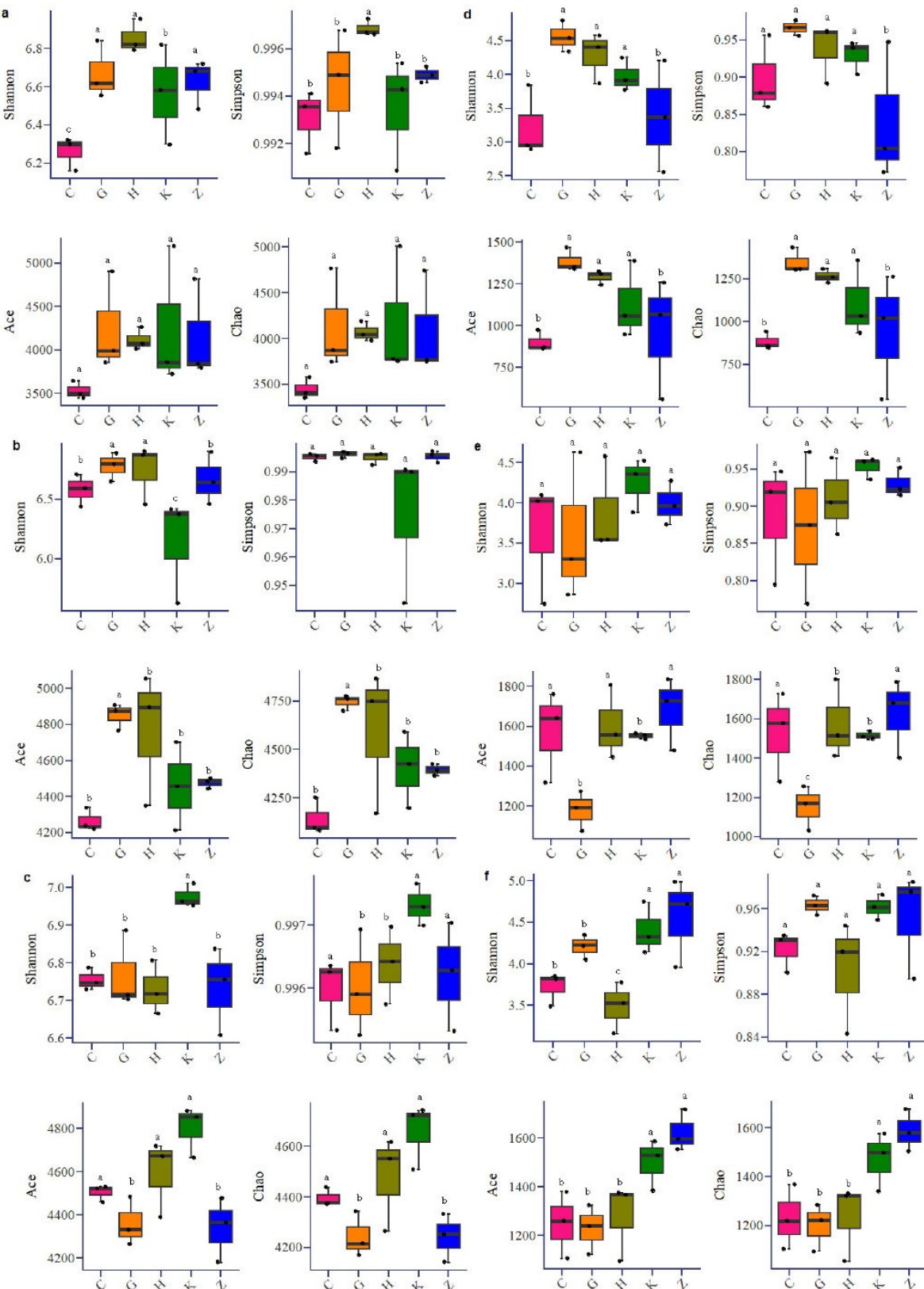

**FIG 2** Alpha diversity of bacteria and fungi. (a–c) Bacterial DTS, XTS, and ZTS. (d–f) Fungal DTS, XTS, and ZTS. Letters in the figure indicate significance.

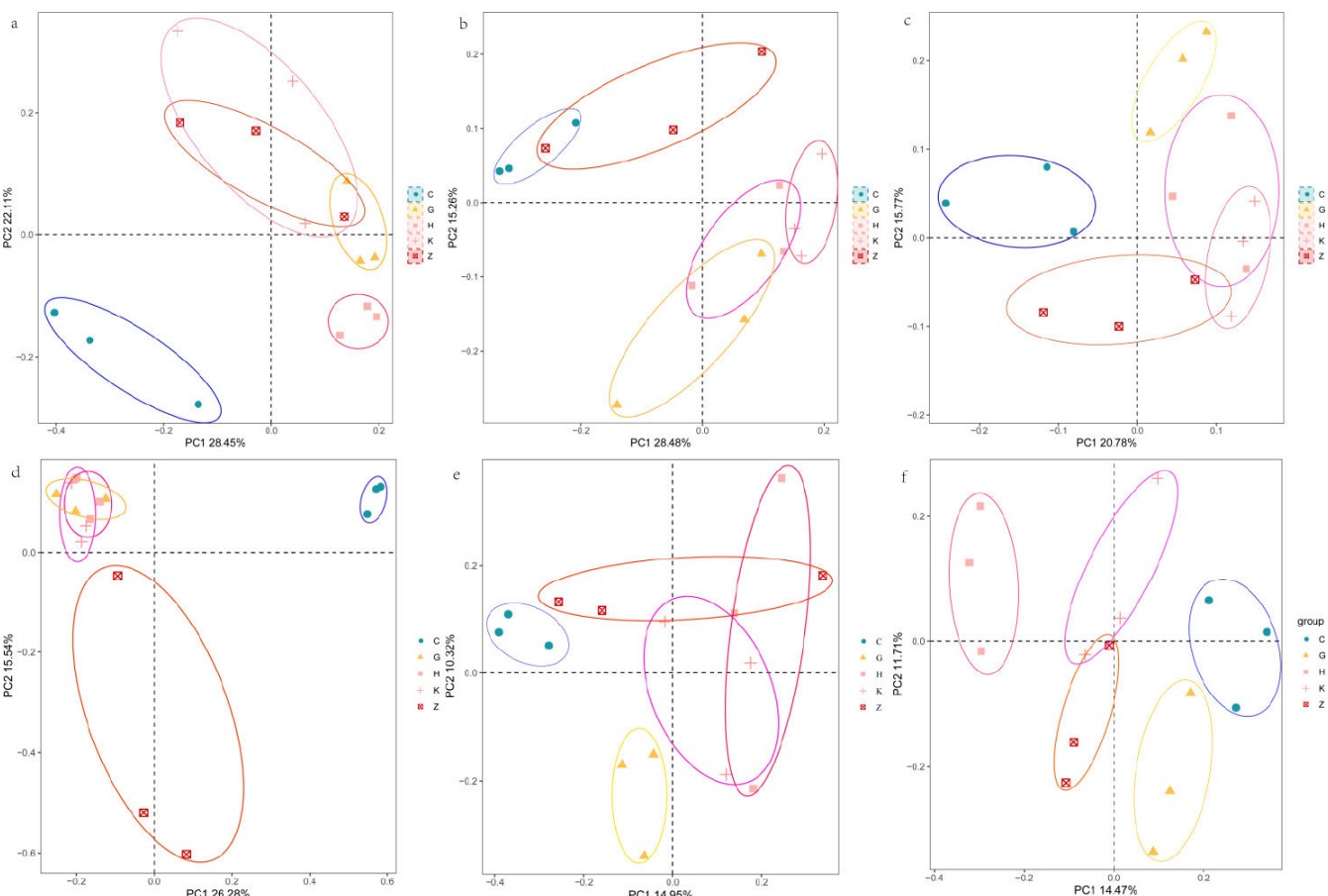

**FIG 3** β-Diversity of bacterial and fungal communities. (a–c) Bacterial DTS, XTS, and ZTS, respectively. (d–f) Fungal DTS, XTS, and ZTS.

Bray-Curtis and unweighted UniFrac differences in operational taxonomic units (OTUs) were compared for soil microbial community structure using principal coordinates analysis (PCoA) to characterize the microbial beta diversity. As shown in Fig. 3, the significant experimental results of analysis of variance using distance matrices (ADONIS) and analysis of similarities (ANOSIM) indicated that there were significant differences in the classification of bacterial and fungal communities among different vegetation types in different regions. Differences in the bacterial community structure among coniferous forests (ADONIS $R = 0.436$, $P = 0.05$), broad-leaved forests (ADONIS $R = 0.456$, $P = 0.05$), mixed forests, and fungi were observed, but the overall variation showed that fungi were smaller than bacteria. There were significant differences in bacteria among the three sampling points in the eastern, central, and western regions ($P = 0.05$), whereas fungi showed significant differences in both broad-leaved and mixed forests ($P = 0.05$).

## Network analysis of soil microbial co-occurrence of different tree species compositions

To comprehensively explore the connections between environmental parameters and microbial networks, environmental parameters were taken as nodes and integrated into one microbial network. The total interdomain network and the extracted subnetworks are shown in Fig. 4. In Fig. 5, the strength of correlation between each node in different vegetation types is further shown.

In Fig. 4, $N$ represents the number of nodes in the network; $E$ represents the edges; and the number of nodes and links in a collinear network is an important property of network stability. In the present study, the overall network complexity was as follows: broad-leaved forest ($N = 82$, $E = 318$) and mixed forest ($N = 85$, $E = 378$) > coniferous

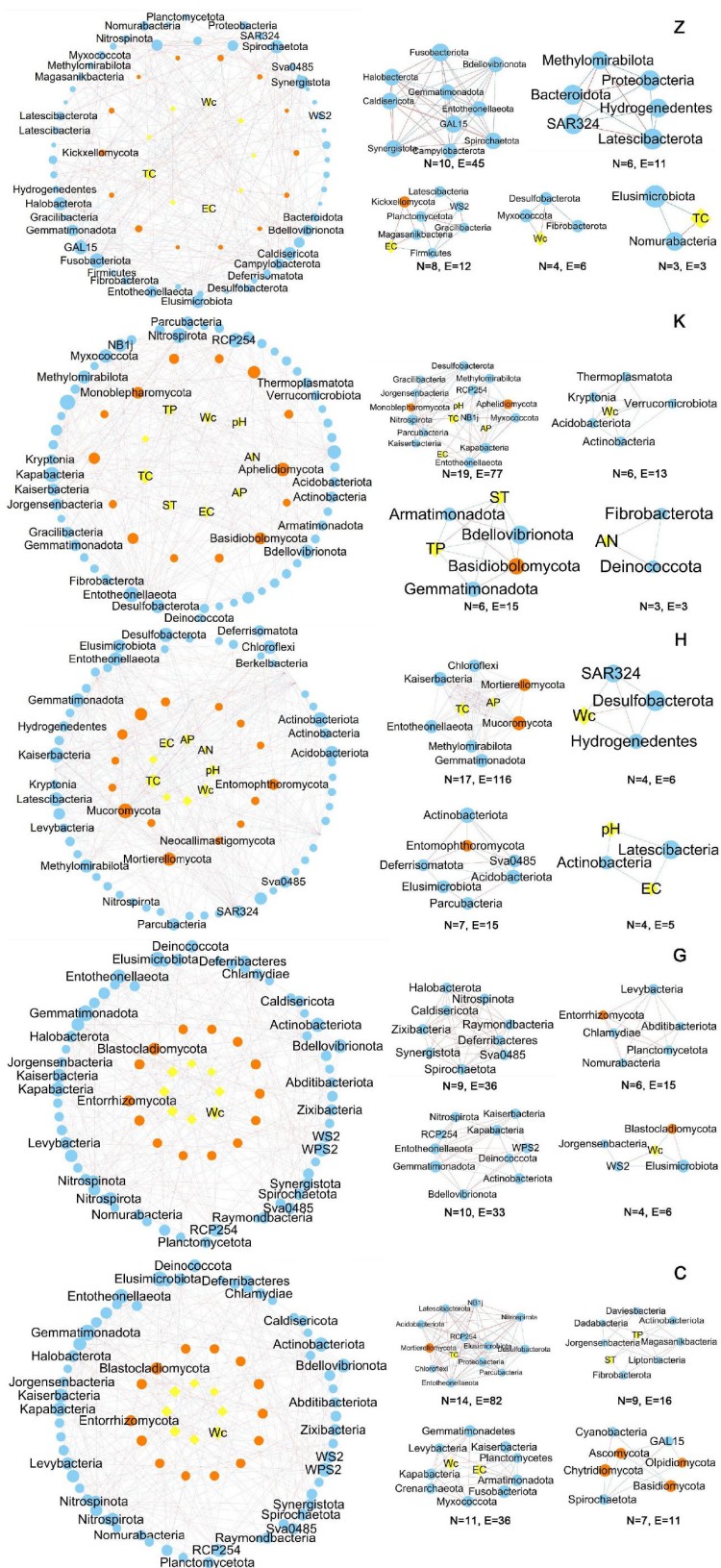

**FIG 4** Network diagram of bacteria-fungus-soil factor domains of different vegetation types (yellow nodes are soil factors; orange nodes are fungal phyla levels; blue nodes are bacterial phylum levels; red lines indicate a positive correlation; and blue lines indicate a negative correlation).

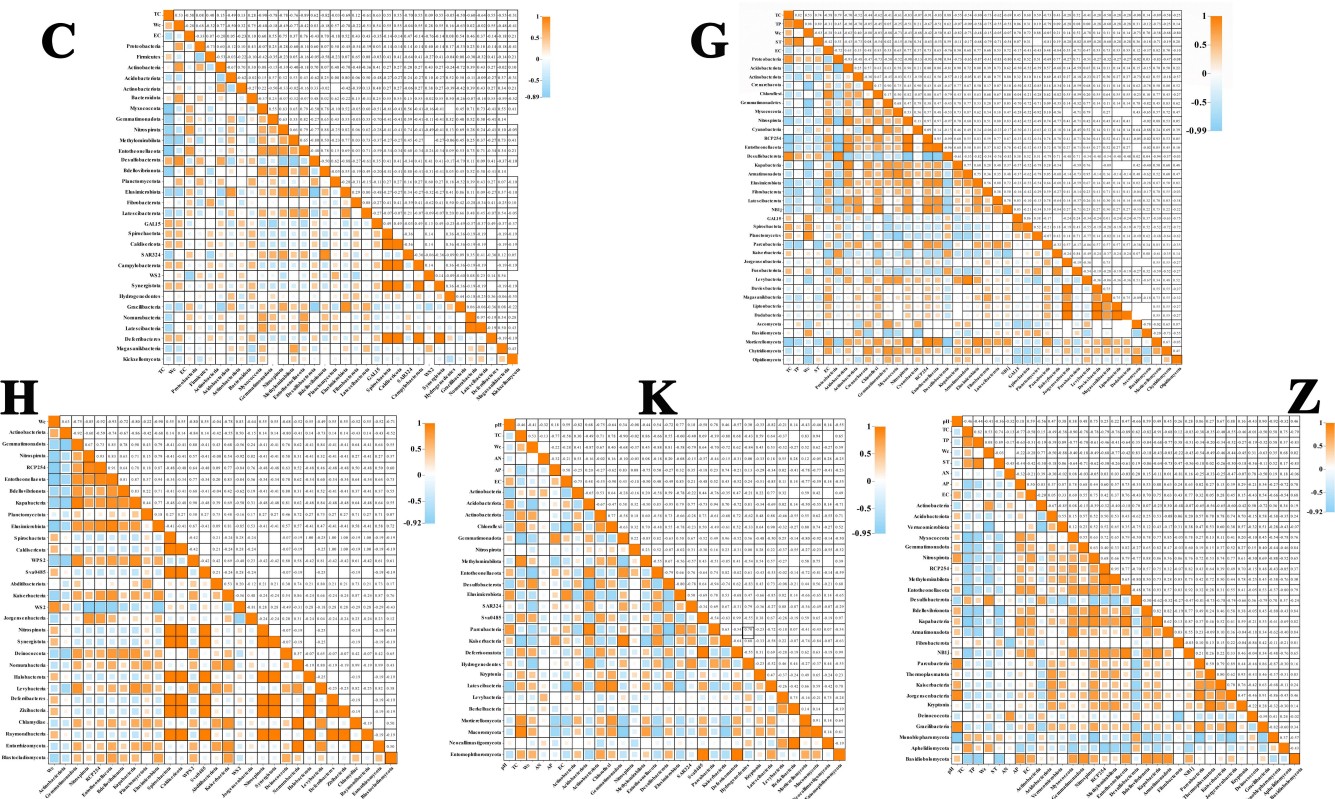

**FIG 5** Strength of correlation between each node in different vegetation types. The number in the upper right corner represents the strength of the correlation. C, herbal; G, shrub forest; H, mixed coniferous and broad-leaved forest; K, *Ulmus pumila* forest; Z, *Picea schrenkiana* forest.

forest ($N$ = 81, $E$ = 281) > shrubs ($N$ = 80, $E$ = 271) and herbs ($N$ = 76, $E$ = 356). The average connectivity and clustering coefficient of the microbial network in the mixed forest soil were higher, and there were more interactions. In contrast, the average path distance and modularity of the coniferous forest network were higher. Surprisingly, the interaction in herbal herbs is quite high.

In co-occurrence networks, higher modularity values may be related to higher resource availability and habitat complexity because modularity reflects the heterogeneity of habitats and different selection mechanisms. Similarly, the trend of the modularity index was mixed forest (0.941) > shrubs (0.932) coniferous forest (0.87) and broad-leaved forest (0.884) > herbs (0.701). Modularity index greater than 0.4 indicates that the network is modular, and unexpectedy, shrubs exhibit a higher level of modularity.

The interdomain network of bacterial-fungi soil factors consisted of 2,706 bacterial nodes, accounting for 72.88% of the total nodes, and 1007 fungal nodes, accounting for 27.12% of the total nodes. The key microbial groups in the network also differed among stands. At the phylum level, Ascomycota and Basidiomycota were the main groups of fungal networks, whereas Proteobacteria, Acidobacteria, and actinomycetes were the main groups of bacterial networks. Most nodes in the network were classified as peripheral; however, the number and classification affiliations of these nodes varied among the five vegetation types. Most hub and connector microorganisms were significantly positively correlated with soil carbon and water content and negatively correlated with N content. Node connectivity is a key topological attribute that describes the degree of connectivity between a node and its adjacent nodes. Compared to interdomain networks of the five vegetation types, among all environmental parameters, total carbon served as the hub with higher connectivity to bacteria. In the five ecosystems, node connectivities were 15, 15, 17, 8, and 16, followed by water content, with connectivities of 11, 10, 9, 10, and 12. The factors with high connectivity to fungi

were total carbon, soil temperature, and water content. Further, as network complexity increases, the proportion of fungi increase, while bacteria decrease. At the same time, the negative links in the network also increase. The dominant fungi and bacteria were evenly distributed in the hardwood forest, indicating the robustness of the network.

Most key bacterial and fungal groups are related to soil carbon and nitrogen metabolism, decomposition of organic compounds, soil fertility, and pathogenicity. In key bacterial genera, actinomycetes collaborate with other bacterial groups to participate in C and energy flows; that is, spore fungi participate in soil nutrient turnover. Among the key fungal genera, *Holtmann* can assimilate C from multiple sources and possesses extracellular enzymes; *Trichophyton* is classified as a plant pathogen (20) and promotes nutrient acquisition. The enrichment of these key soil bacteria and fungi with nutritional functions may affect the microbial network, change the soil nutrient cycle and transformation, and ultimately promote the absorption and utilization of plant nutrients in mixed forests.

## Prediction of soil microbial functions in different tree species compositions

FAPROTAX and PICRUSt are commonly used to predict the potential functions of bacteria by comparing gene sequences from different databases (21). The comparison database of PICRUSt is Greengenes, which is relatively complete and accurate and has been applied in the functional analysis of soil microorganisms (22, 23). As shown in Fig. 6, functional prediction obtained six primary functional layers: environmental information processing, metabolism, genetic information processing, organic systems, human diseases, and cellular processes. There are 35 secondary functional layers with abundant soil bacterial functions. Among the different vegetation types, the main functions of forests are genetic information processing; shrubs are associated with organic systems and metabolism, while herbs are related to environmental information processing and cellular processes. The soil bacterial community in the East Tianshan Mountains was active in genetic information processing and organic systems, whereas the soil microorganisms in the West Tianshan Mountains had a high abundance of functional genes involved in environmental information processing and metabolism.

To explore the relationship between bacteria and nutrient cycling, FAPROTAX analysis was conducted, and the results showed that there were 45 related functional pathways, of which 27 were related to carbon (C) and nitrogen (N) cycling. In general, bacteria involved in the C cycle were the most abundant, accounting for approximately 5%–60%. Genes related to the carbon cycle were significantly different between the *Picea schrenkiana* and *Ulmus pumila* forests ($P = 0.001$), and there were many carbon cycle pathways in broad-leaved forests. In coniferous forests, functional bacteria related to the C cycle are more abundant and are strongly related to Proteobacteria, actinomycetes, and Acidobacteria. Among the other elements, nitrification and ammonia oxidation were stronger in broad-leaved forest soils; nitrate reduction was stronger in coniferous forests; and manganese oxidation was stronger in mixed forests. The C-cycle process, especially the cellulose metabolism pathway, significantly increased in the Western Tianshan Mountains. In the Eastern Tianshan Mountains, there were abundant functional groups related to the biosynthesis of nitrogen-related secondary metabolites ($P = 0.01$). In contrast, in the Central Tianshan Mountains, the plant pathway and chitin decomposition functions were stronger.

Based on the FUNGuild database for the functional prediction of fungi, nine primary functional layers were obtained: pathotroph, saprotroph, symbolotroph, pathotroph saprotroph, pathotroph symbolotroph, saprotroph symbolotroph, pathotroph saprotroph symbolotroph, and pathogen saprotroph symbolotroph, and 26 secondary functional layers. Owing to the gradual decrease in soil nutrition from Western Tianshan to the Eastern Tianshan Mountains, the richness of functional genes was characterized by XTS > ZTS > DTS. Ectomycorrhizae are dominant in coniferous forests, whereas orchid and fungal mycorrhizae are dominant in broad-leaved forests. Some pathogenic fungi

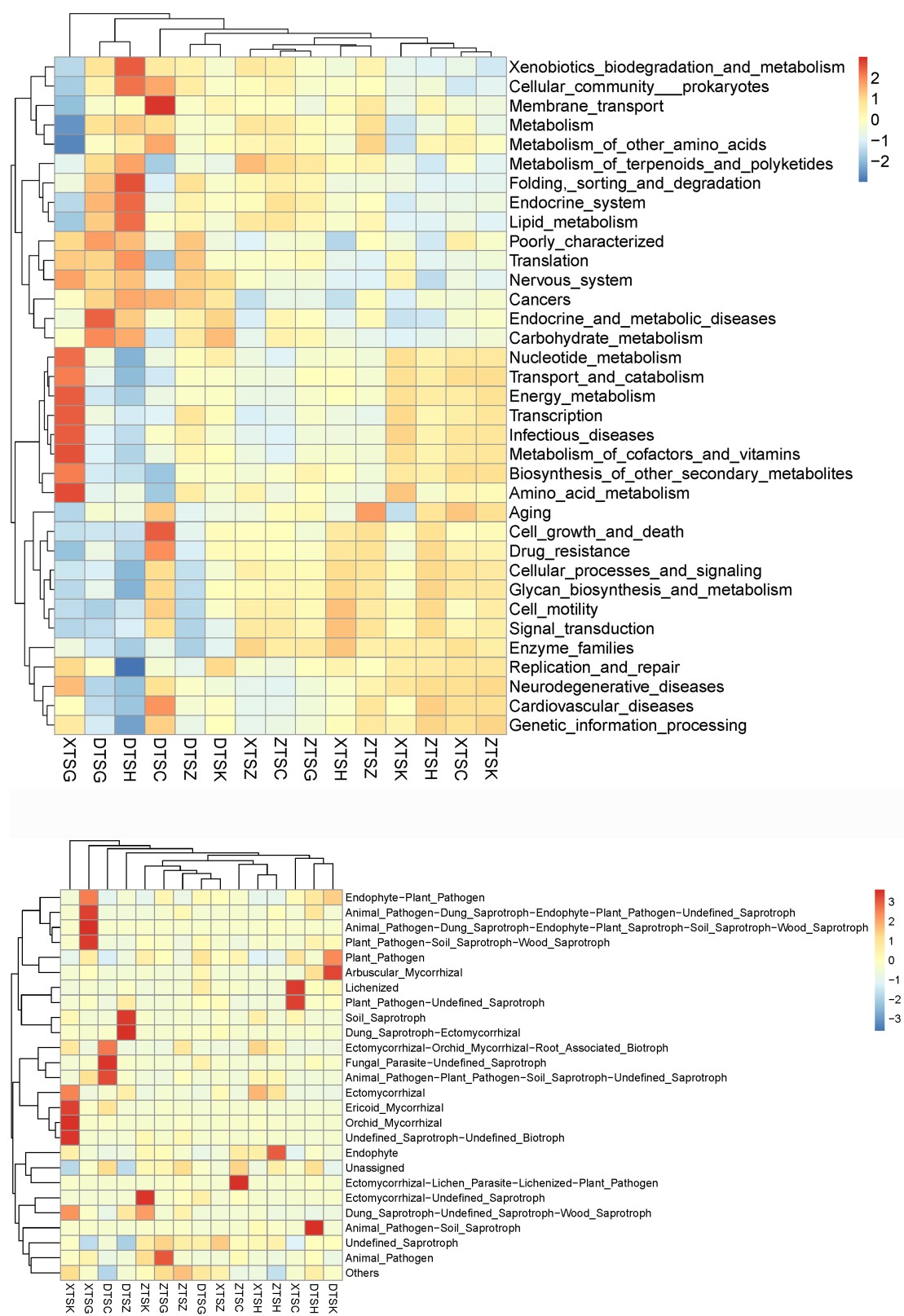

**FIG 6** Heatmap of functional gene prediction in bacteria (top) and fungi (bottom).

were present, whereas mixed forests exhibited both. Shrubs have various functions, whereas herbs are mostly lichens because of their low water content.

## DISCUSSION

### Soil physicochemical factors of different vegetation types

The soil pH of coniferous forests is lower than that of broad leaf forests, whereas the pH of mixed forests falls between the two. This is because conifers tend to have lower soil pH and cation concentrations (7). *Picea schrenkiana* reduces the pH value of the soil by increasing the number of anions in the soil solution, the amount of acid infiltrated into the soil, the degree of protonation of the soil acid, and the strength of the soil acid. Nutrients, such as total carbon and phosphorus, are higher in mixed forests compared to pure forests. This is because mixed forests have a higher average growth rate than single-species forests, resulting in the accumulation of more organic matter over time. The results of water content and soil temperature indicate that the western region exhibits the most favorable thermal conditions. This is attributed to lower precipitation levels and a lower annual average temperature in the western area.

### Soil microbial composition and diversity in different vegetation types

In the present study, the species and abundance of bacteria in herbaceous plots were higher, whereas fungi were more abundant in forests, especially in pure coniferous forests. This is because the roots and litter of herbaceous and shrubby plants have the characteristics of rapid replacement and decomposition compared to those of trees. Herbal roots are the main carbon source for bacteria, whereas arbor and shrub roots are the main carbon sources for fungi. Moreover, arbor roots can form a symbiotic relationship with ectomycorrhizal fungi and affect the soil fungal community through nutrient exchange or hyphae (24), thereby exerting a stronger driving effect on the soil fungal community. Compared with grasslands, the dominant bacterial phyla in the forest changed from Proteobacteria, which prefer a nutrient-rich soil environment (25), to actinomycetes, and the dominant fungal phyla changed from Ascomycota to Basidiomycota. These findings are consistent with the copiotrophic/oligotrophic bacterial hypothesis proposed by Fierer et al. (26). The higher abundance of Firmicutes in spruce stands is due to their lignin-degrading activity, which is crucial for lignin decomposition.

Members of Proteobacteria and Acidobacteria are almost ubiquitous in these soil types, and their advantages are more common in coniferous forest soils. Both can decompose cellulose, and it is relatively difficult to obtain the required nutrient resources through the weathering of hard, decomposed organic matter or minerals (27, 28). Members of actinomycetes have the ability to turnover organic matter, degrade recalcitrant molecules (29), and exhibit high plasticity in C utilization. They can utilize fresh substrates (including cellulose) or polycyclic aromatic compounds to decompose polysaccharides and phenolic compounds in the litter, with a greater advantage in broad-leaved forests (30).

Ascomycota are the most abundant fungal species in forest soils, promoting carbon transformation and participating in the metabolism of refractory organic macromolecules (31). Basidiomycetes and Mortida are associated with the process of soil carbon transformation, exhibiting a strong ability to decompose lignocellulose in plant residues (32, 33). Additionally, they can inhibit the growth of other fungi by competing for carbon resources. Generally, Basidiomycota members prefer rich resources and high plant species richness compared to Ascomycota members, so their abundance is higher in mixed forests and broad-leaved forests (34). Similarly, heterotrophic Bacteroidetes and Firmicutes are associated with the decomposition of soil organic matter, with a higher abundance in mixed forests rich in organic matter (35). The higher nitrogen concentration in the soil of the elm forest increased the relative abundance of co-trophic groups (Bacteroidea and Firmicutes) and decreased the relative abundance of oligotrophic groups.

The results of beta diversity analysis show that vegetation type explains fungal differences more than location, while in bacteria, the opposite is true. On the one hand, bacteria are sensitive to pH; bacteria in coniferous forests differ greatly from those in other vegetations. On the other hand, because the fungi produce hyphae and the movement distance increases, their heterogeneity is not sensitive to soil factors and is greatly affected by vegetation type.

## Microbial-soil co-occurrence networks in soil of different vegetation types

Analyzing the interactions between microorganisms and the soil environment using ecological networks can provide new insights into the mechanisms of community stability. Previous studies on microbial co-occurrence networks have mainly focused on fungi (36) and bacteria (37, 38) in ecosystems. Only a few studies have included soil physicochemical factors in the co-occurrence network to reveal the potential internal relationships between soil and microorganisms.

Except for shrublands, EC, total carbon (TC), and water content (Wc) as the connector node appeared in other networks, indicating that soil salinity and water stress in the Xinjiang arid zone had a significant impact on microorganisms (39). Drought and high temperature exceeding the threshold will cause significant pressure on microorganisms and affect the stability of microbial network (40, 41), and since Xinjiang is in an arid region, microbial processes are more sensitive to changes in soil moisture, soil water content, soil temperature, and EC in the soil microbial network and often become key factors.

Complex networks indicate that most microbial species share similar ecological niches, whereas microbes in simple networks have variable ecological niches. In this study, the number of key groups in forest soil was significantly higher than that in shrub and herbaceous soils, and the presence of more key groups implied higher network stability. In the soil of coniferous and mixed forests, the number of symbiotic network modules of bacteria and fungi increased significantly, indicating that microorganisms in these two systems had a higher degree of cooperation and communication, which is consistent with previous research on forest soil (15). Furthermore, compared to the single forest, the mixed forest increased the number of negative links in the bacterial group network, indicating that competitive interactions in the bacterial network increased. In contrast, there were more cooperative relationships between the fungal groups. The key OTUs in the broad-leaved forests were mainly oligotrophic, showing a negative correlation with organic matter. This was due to the abundance of microorganisms related to carbon degradation, which resulted in lower accumulation of organic matter. This confirmed the conclusion of functional gene prediction. In the mixed forests, the interaction between bacteria and fungi increased, indicating that mixing promoted microbial activity.

The numbers of nodes and links and modularity are the key attributes of the network (37, 38). The interdomain network constructed by mixed forest has higher numbers of nodes, edges, and module degree, indicating that the coupling enhances the functional microbial group and enhances the ability to resist environmental stress by improving the connectivity of bacterial and fungal communities (42). In this study, the proportion of positive interaction edges was higher between hardwood forest (58.71%) and mixed forest (55.42%) networks, indicating that bacterial and fungal communities form tight organizations through cooperation, thus enhancing the complexity of community structure and ecosystem stability. This can be explained by a reduced competition between species, related to nutrient resources, and microorganisms may cooperate to adapt to similar niches (12). Modules represent functional groups of species; taxa belonging to the same module have similar niches; and higher modularity favors resistance to environmental changes. In this study, the modularity value of bacterial and fungal co-occurrence networks in all sample loci was greater than 0.4, and the modularity degree of bacterial network in woodland and grassland soil was lower than that of mixed forest soil, and the low modularity degree of herbaceous soil network

may be due to low soil water content, which is less likely to stimulate microbial niche differentiation (43).

The Aciidobacteria, Actinobacteria and Proteobacteria, which dominate the co-occurrence network, are closely associated with soil C content storage and alter the ability of microorganisms to decompose soil organic matter by regulating the production of extracellular catabolic enzymes. Proteobacteria are often the dominant nitrogen-fixing bacteria in soil ecosystems and play key roles in organic matter decomposition and nutrient cycling by connecting other microbial members in the network. Actinomycetes exhibit hyphal growth patterns in soil, enabling plants to expand their surface area to deeper soil layers to take up nutrients; these mycelia form soil aggregates and act as active components for preserving water and nutrients. The dominant fungal phylum in soil is Ascomycota and Basidiomycota, which usually form symbionts with roots in dry and cold habitats (44). Most saprophytic fungi are Ascomycota that rapidly metabolize organic substrates in soils and which are more involved in cellulolysis compared to Basidiomycetes (45).

The key groups exhibited different response modes for different soil factors. Proteobacteria/Acidobacteria usually increase with an increase in C (46, 47) and soil water content (48). Actinomycetes were connected to TC and TN at the nodes. Basidiomycota and Ascomycota were positively correlated with the TC and TN levels. Dependence on soil pH is not unique to acidic bacilli. In this study, the relative abundances of Actinobacteria, Bacteroides, and Bacillus were also significantly related to soil pH, further indicating that at the sample plot scale, soil chemistry is an important determinant of the bacterial community (49).

## Functions of microbes differ under different vegetation types

There are differences in the relative abundance of various functional genes in different microbial community structures, and predicting functional genes to some extent can reflect the interaction, regulation, and adaptability of soil microbial communities underground (50). These results indicate that the vegetation type has a significant impact on the function of microorganisms.

This study confirmed that a mixture of coniferous and broad-leaved species promoted the functional gene abundance of specific microbial groups (51); and the soil microbial functional diversity was higher in broad-leaved forests with a loose soil structure, low carbon-nitrogen ratio, and high total nitrogen and available nitrogen concentrations. Zhang et al. (52) found that microbial functions shifted from information processing and storage in forests to metabolism in shrubs, as demonstrated in this study. The functional characteristics of microbial communities can affect the structure and diversity of bacterial communities. The high abundance of metabolic and organic system functional genes in shrublands enhances soil bacterial metabolism and enhances the diversity of bacterial community structure. The main function of functional gene metabolism is to absorb nutrients from the soil, thereby ensuring rapid bacterial survival (53). Mixed forests absorb more nucleotides and amino acids from the soil via these functional genes, thereby improving bacterial diversity.

Some fungi in the soil participate in nutrient cycling, whereas others exist as pathogenic bacteria (54). Compared to broad-leaved and mixed forests, a decrease of nutrient in soil environment led to the decrease of symbiotic fungi abundance in coniferous forests (55). The increase in the abundance of saprophytes is because saprophytes, as decomposers of organic matter, regulate the flow of soil carbon and nitrogen, and can retain and redistribute nutrients in the mycelium, thus overcoming local nutrient constraints (56, 57). Conifer species planted in soil contain high cellulose and low lignin contents (58), which are conducive to the differentiation of saprophytic fungi and ectomycorrhizal communities (59). Owing to the water limitation of herbs, the proportions of pathogenic saprophytic bacteria, parasitic saprophytic bacteria, and endophytic saprophytic bacterial functions increase, which is consistent with the relative abundance increase of ascomycetes under drought stress (60).

The Tianshan Mountains are located in an extremely arid environment. However, because of the adequate precipitation within some of the mountainous regions, there is a zonal forest area within the mountains. Although the area of this forest area is small, it is distinctly different from the background ecological conditions and is important in supporting the existence of habitats for flora and fauna in the arid region. In this context, this study investigates the mechanisms of microbial-soil interactions in natural forest stands in the arid mountainous region, which is useful for enriching the study of ecological processes in forest ecosystems. Due to the limitation of the distribution of natural vegetation species in the Tianshan Mountains, this work focuses only on the most typical *Picea schrenkiana* forests and *Ulmus pumila* forests for sampling and investigation; future work can be carried out to study mixed forests including other types of mixed forests, such as coniferous mixed forests and broad-leaved mixed forests.

## MATERIALS AND METHODS

### Study setting

The Tianshan Mountains are located in the central region of Asia in an east-west direction, with a total length of approximately 4,000 km and a north-south width of approximately 150 km. The research area focuses on the northern slope of the Tianshan Mountains. The vegetation samples were collected from the subalpine zone of the mountain forest, predominantly consisting of *Picea schrenkiana* and *Ulmus pumila*. The soil type found beneath the forest was mountain gray-brown forest soil. The shrubs were mainly *Rosa platyacantha* Schrenk and *Berberis atrocarpa*. The mountainous grassland plot is located in the middle mountainous area, with vegetation mainly consisting of understory plants such as *Stipa capillata*, *Achillea millefolium*, *Geranium*, and *Fragaria vesca*. The three sample plots in this study were located at the Balikun Forest Farm (Eastern Tianshan Mountains, DTS, 92°55′56–93°44′27E, 43°19′21–43°36′14N), Haxionggou Forest Park in Urumqi (Central Tianshan Mountains, ZTS, 87°56′6″–87°59′2″E, 43°38′44–43°51′12N), and Wusu Foshan Forest Park (Western Tianshan Mountains, XTS, 84°35′52–84°37′38E, 44°7′32–45°8′18N).

The research site is located in Tianshan Mountain, Xinjiang, China (84°35′52.2312″E–93°44′38.2956″E, 43°35′45.60″N–43°56′28.68″N), belonging to temperate continental climate, with annual average temperature and precipitation of 6.51°C and 184.88 mm, respectively. This research site included five typical types of Tianshan forests: *Picea schrenkiana* forest, *Ulmus pumila* forest, mixed coniferous and broad-leaved forest, shrub forest, and grassland plot, with minimal human interference (e.g., harvest and fire). Among the five vegetation types, a total of 15 sample plots were established, each measuring 400 m$^2$ (20 m × 20 m), with three repeated samplings for each treatment. This resulted in a total of 45 soil samples being collected for testing. In each plot, soil sampling was conducted during the peak season of plant growth, specifically from June to August 2022. To ensure representative sampling, a five-point sampling method was employed, targeting the 0- to 20-cm soil layer of the forest. It was important to avoid oversampling specific locations, such as tree roots.

### Soil trait determination

After removing the visible root and plant residues, each soil sample was divided into two subsamples. One sample was air-dried and sieved through a 2-mm mesh for chemical analysis, while the other sample was stored at −80°C for DNA extraction. The soil moisture content (Wc) was measured by drying the soil samples for 8 hours at 105°C.The soil temperature (ST) was measured using a soil thermometer; the pH was measured using the potentiometer method; soil conductivity (EC) was determined using a conductivity meter; TC analysis was conducted using the potassium dichromate external heating method. Total phosphorus (TP) and available phosphorus (AP) analysis was performed using the sodium bicarbonate leaching molybdenum antimony

resistance colorimetric method. Total nitrogen(TN) and available nitrogen(AN) were determined using the semi-micro Kjeldahl method. Soil TP, TC, TN, AN, and AP were measured using a UV spectrophotometer.

Microbiological determination was conducted using high-throughput sequencing. The genomic DNA of the sample was extracted using the SDS method, and the purity and concentration of DNA were assessed through agarose gel electrophoresis. Specific primers with barcode were selected based on the sequencing regions, and Phusion from New England Biolabs High Fidelity PCR Master Mix with GC Buffer and efficient high-fidelity enzymes were used for PCR to ensure efficient and accurate amplification. The PCR products were then analyzed by electrophoresis using a 2% agarose gel. Qualified PCR products were purified by magnetic beads, quantified by enzyme labeling, and mixed in equal amounts based on their concentrations. After thorough mixing, the PCR products were again analyzed using 2% agarose gel electrophoresis.

The sequencing machine data were stitched and quality controlled, and the chimera filtering was performed to obtain valid data that can be used for subsequent analysis. To study the species composition of each sample, OTUs were clustered with 97% identity (Identity) for effective tags for all samples, and then the sequence of OTUs was species annotated. 16S uses Silva database; the annotation method is mothur method. During the annotation process, the threshold is set to 0.8–1.0; that is, the annotation results with confidence higher than the set threshold can be fully output. The species annotation of ITS is the Unite database (unite + INSD), annotated using the Blast method. The whole process is performed in Qiime (v.1.9.1) (http://qiime.org/scripts/assign_taxonomy.html).

### *Primer-corresponding regions*

The 16S V4 region primers (515F and 806R) were used to identify bacterial diversity, and ITS1 primers (ITS5-1737F and ITS2-2043R) were employed for fungal diversity analysis.

### Data analysis

One-way analysis of variance was used to determine the differences in soil physico-chemical factors, relative abundance of bacterial and fungal communities, and the Shannon diversity index under different vegetation types using SPSS (v.23.0) ($P = 0.05$). A non-metric multidimensional scale based on Bray-Curtis distance and the ADONIS test were used to evaluate the beta diversity of the bacterial and fungal communities, and analysis of variance was conducted. The soil microbial interdomain network was constructed based on the Pearson correlation analysis, and the above calculations were completed using R (v.4.2.1). The microbial-soil network was visualized using Cytoscape. The microbial function was predicted using FAPROTAX and PICRUSt analyses.

### Conclusion

This study showed that different vegetation types in the Tianshan Mountains had an effect on soil characteristics, soil microbial community composition, and functional genes. The fungal community structure is mainly affected by the tree species, while the bacterial community structure is mainly affected by soil carbon. Among the different types, the mixed forest had the best soil nutrient and hydrothermal conditions, followed by the broad-leaved forest, and the Yunshan forest was colder, lacking nutrients, and having a lower pH due to its higher altitude. With the increase of longitude from west to east, the soil hydrothermal and nutrient conditions gradually turned barren, and the physical and chemical characteristics of each sample point in the East Tianshan Mountains varied little. The results of microbial composition showed that a total of 75 phyla were found in bacteria and 19 fungi. Among the types, the dominant bacteria were *Proteus* (27%–41%), actinomycetes (9.17%–11.83%), and the dominant fungi were ascomycetes (50.22–63.95%) and Basidiomycetes (8.99–28.09%). The fungal community structure of the mixed forest was more similar to that of the coniferous forest (19), while the bacterial community structure was more similar to that of broad-leaved plants.

The significant experimental results of ADONIS and ANOSIM showed that there were significant differences in the classification of bacterial and fungal communities between different vegetation types in different regions. The co-occurrence relationship between soil and microbial domains showed that the average connectivity (812.50) and clustering coefficient (8.89) of microbial networks in mixed forest were high, and there were many interactions. The average path distance (3.17) and modularity (0.87) of the taiga network were higher. The results of this study provide a reference for the correlation between microorganisms and soil factors in the Tianshan Mountains and are of great significance for the construction of interdomain networks of different vegetation types. This study is limited by the causes of vegetation distribution in Tianshan Mountains, and only the mixed forest sampling of coniferous forest and broad-leaved forest, allowing multiple mixed forest studies in the future. Future work could alter or increase the stand species and look at assessing the effect of tree species type on the dominant species.

## ACKNOWLEDGMENTS

This work was supported by The third comprehensive scientific investigation project in Xinjiang (2021XJKK0900) and the Key Laboratory of Oasis Ecology.

## AUTHOR AFFILIATIONS

[1]College of Ecology and Environment, Xinjiang University, Urumqi, China
[2]Key Laboratory of Oasis Ecology, Ministry of Education, Urumqi, China
[3]Xinjiang Jinghe Observation and Research Station of Temperate Desert Ecosystem, Ministry of Education, Jinghe, China

## AUTHOR ORCIDs

Qian Guo  http://orcid.org/0000-0002-4025-393X
Lu Gong  http://orcid.org/0000-0001-8447-9052

## FUNDING

| Funder | Grant(s) | Author(s) |
| --- | --- | --- |
| the third comprehensive scientific investigtion project in xinjiang | 2021XJKK0900 | Lu Gong |

## AUTHOR CONTRIBUTIONS

Qian Guo, Conceptualization, Data curation, Formal analysis, Investigation, Writing – original draft | Lu Gong, Conceptualization, Funding acquisition, Investigation, Methodology, Writing – review and editing

## DATA AVAILABILITY

The data used in this paper are high-throughput sequencing data, stored in the National Center for Biotechnology Information (PRJNA989514).

## ADDITIONAL FILES

The following material is available online.

Open Peer Review

**PEER REVIEW HISTORY (review-history.pdf).** An accounting of the reviewer comments and feedback.

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
