## [Reviewer comments · Microbiology Spectrum]

Microbiology Spectrum

Compared with pure forest, mixed forest alter microbial diversity and increase the complexity of inter-domain networks in arid areas

Guo Qian and Lu Gong

Corresponding Author(s): Lu Gong, Xinjiang University

Review Timeline:

Submission Date:	June 30, 2023
Editorial Decision:	September 8, 2023
Revision Received:	October 19, 2023
Editorial Decision:	November 8, 2023
Revision Received:	November 21, 2023
Accepted:	November 22, 2023

Editor: Diyan Li

Reviewer(s): Disclosure of reviewer identity is with reference to reviewer comments included in decision letter(s). The following individuals involved in review of your submission have agreed to reveal their identity: Muhammad Afzal (Reviewer #1); Basanta Kumar Biswal (Reviewer #7)

Transaction Report:

DOI: <https://doi.org/10.1128/spectrum.02642-23>

September 8, 2023

Dr. lu gong
Xinjiang University
urimqi
China

Re: Spectrum02642-23 (Compared with pure forest,mixed forests alter microbial diversity and increase the complexity of inter-domain networks in arid areas)

Dear Dr. lu gong:

Link Not Available

Sincerely,

Diyang Li

Journals Department
Reviewer comments:

Reviewer #1 (Comments for the Author):

A manuscript titled "Compared with pure forest, mixed forests alter microbial diversity and increase the complexity of inter-domain networks in arid areas" studied the soil microbial community structure of five typical vegetation types in Tianshan, Xinjiang, China, using bacterial 16S rRNA and fungal ITS sequences. It has been stated that bacterial community is more related to soil C and fungal community to forest type. Overall manuscript is well-written and the conclusion is supported by the data. My comments on the manuscript and supporting information document are detailed below.

General comments:

1. Introduction needs a paragraph where the microbial role is explained for soil processes such as degradation, mineralization and transformation.

2. Material and methods for soil parameters need to be explained in more detail with the instruments being used. Microbial data analysis also needs to be explained in more detail with the pipeline used for analysis and taxonomy annotation databases.
3. In the result section, figures need a little more explanation.
4. Discussion needs to explain with the key microbial process performed by the dominant fungal and bacterial phyla in the soil in the current study.

Specific comments

Introduction

1. Table 1. Here are two times removed. Is given in the very beginning. Place it properly somewhere in the main text.
2. Many typographical and spacing errors. L1. (forest,mixed), L63 (2013)have), L 66 (properties(Xue) and so on.....

M & M:

3. L103-105, L120-122 (Give sampling Coordinates). Re-write this sentence.
4. L124. Scientific names should be italics. (apply throughout the MS)

Reviewer #3 (Comments for the Author):

Comments, suggestions and questions are described in paper for the author.
It's important that figure captions with descriptive content should be added to the paper.

Reviewer #5 (Comments for the Author):

This article has ignored the microbial interactions. So it focused on bacteria and fungal community as an individual factors. However in conclusion, they wrote that " Our research further elucidates the factors that affect the structure of microbial communities and can be applied to forest management, ecosystem processes, and ecosystem services."

Staff Comments:

Preparing Revision Guidelines

Please return the manuscript within 60 days; if you cannot complete the modification within this time period, please contact me. If you do not wish to modify the manuscript and prefer to submit it to another journal, please notify me of your decision immediately so that the manuscript may be formally withdrawn from consideration by Microbiology Spectrum.

A manuscript titled “Compared with pure forest, mixed forests alter microbial diversity and increase the complexity of inter-domain networks in arid areas” studied the soil microbial community structure of five typical vegetation types in Tianshan, Xinjiang, China, using bacterial 16S rRNA and fungal ITS sequences. It has been stated that bacterial community is more related to soil C and fungal community to forest type. Overall manuscript is well-written and the conclusion is supported by the data. My comments on the manuscript and supporting information document are detailed below.

General comments:

1. Introduction needs a paragraph where the microbial role is explained for soil processes such as degradation, mineralization and transformation.
2. Material and methods for soil parameters need to be explained in more detail with the instruments being used. Microbial data analysis also needs to be explained in more detail with the pipeline used for analysis and taxonomy annotation databases.
3. In the result section, figures need a little more explanation.
4. Discussion needs to explain with the key microbial process performed by the dominant fungal and bacterial phyla in the soil in the current study.

Specific comments

Introduction

1. Table 1. Here are two times removed. One of the tables is given in the very beginning. Place it properly somewhere in the main text.
2. Many typographical and spacing errors. L1. (forest,mixed), L63 (2013)have), L 66 (properties(Xue) and so on.....

M & M:

3. L103-105, L120-122 (Give sampling Coordinates). Re-write this sentence.
4. L124. Scientific names should be italics. (apply throughout the MS)

Dear Editor, Dear reviewers

Thank you for your letter. I really appreciate all your comments and suggestions! These suggestions have enabled us to improve our work. Based on the instructions provided in your letter, we uploaded the file of the revised manuscript. Accordingly, we have uploaded a copy of the original manuscript with all the changes highlighted by using the track changes mode in MS Word. Appended to this letter is our point-by-point response to the comments raised by the reviewers. The comments are reproduced and our responses are given directly afterward in a different color (yellow). We would like also to thank you for allowing us to resubmit a revised copy of the manuscript. Thanks again!

Reviewer comments:

Reviewer #1 (Comments for the Author):

A manuscript titled "Compared with pure forest, mixed forests alter microbial diversity and increase the complexity of inter-domain networks in arid areas" studied the soil microbial community structure of five typical vegetation types in Tianshan, Xinjiang, China, using bacterial 16S rRNA and fungal ITS sequences. It has been stated that bacterial community is more related to soil C and fungal community to forest type. Overall manuscript is well-written and the conclusion is supported by the data. My comments on the manuscript and supporting information document are detailed below.

General comments:

1. Introduction needs a paragraph where the microbial role is explained for soil processes such as degradation, mineralization and transformation.

Answer: We thank the reviewer for the indicating this crucial point. We have added one more paragraph about the microbial role from the text Line 78 to Line 90.

2. Material and methods for soil parameters need to be explained in more detail with the instruments being used. Microbial data analysis also needs to be explained in more detail with the pipeline used for analysis and taxonomy annotation databases.

Answer: Thanks for such important suggestion. Specific methods have been added to the text, in L160 to L167.

3. In the result section, figures need a little more explanation.

Answer: Thanks for highlighting this point. We have added more explanatory content in the part of the result section.

4. Discussion needs to explain with the key microbial process performed by the dominant fungal and bacterial phyla in the soil in the current study.

Answer: We thank the reviewer for the suggestion. Has been added in the discussion section, in L443 to L469.

Specific comments

Introduction

1. Table 1. Here are two times removed. Is given in the very beginning. Place it properly somewhere in the main text.

Answer: Thanks for this suggestion. The table 1 has been placed appropriately.

2. Many typographical and spacing errors. L1. (forest,mixed), L63 (2013)have), L 66 (properties(Xue) and so on.....

M & M:

Answer: We are thankful to the reviewer for this suggestion. We have done several format revisions of the full text.

3. L103-105, L120-122 (Give sampling Coordinates). Re-write this sentence.

Answer: Thank you very much for this important suggestion. We have re-written the sentence, and added more detailed information about the sampling sites in the new text Line 126 – Line 129. For the former manuscript L 103-105, it refers to the range of Tianshan Mountains, which is a potentially disputed issue. Given this, we tend to give no coordinates about it.

4. L124. Scientific names should be italics. (apply throughout the MS)

Answer: We thank the reviewer for highlighting this point. All species names have been changed to Italics in the full text.

Reviewer #3 (Comments for the Author):

Comments, suggestions and questions are described in paper for the author.

It's important that figure captions with descriptive content should be added to the paper.

Answer: In the light of your suggestion, the descriptive content in figure captions have been added in the text.

Reviewer #5 (Comments for the Author):

This article has ignored the microbial interactions. So it focused on bacteria and fungal community as an individual factors. However in conclusion, they wrote that " Our research further elucidates the factors that affect the structure of microbial communities and can be applied to forest management, ecosystem processes, and ecosystem services."

Answer: Thanks for such an insightful suggestion. We have revised the content changed the sentence to "This study further investigated the mechanism of interaction between microorganisms and soil in natural stands, which has positive implications for exploring the ecological processes of forest ecosystems." In L511.

Re: Spectrum02642-23R1 (Compared with pure forest, mixed forest alter microbial diversity and increase the complexity of inter-domain networks in arid areas)

Dear Dr. Lu Gong:

Thank you for the privilege of reviewing your work. Below you will find my comments, instructions from the Spectrum editorial office, and the reviewer comments.

Revision Guidelines

Sincerely,
Diyan Li
Editor
Microbiology Spectrum

Reviewer #7 (Comments for the Author):

In this study, authors have investigated the impacts of pure forest and mixed forest types on the soil microbial dynamics. Overall, this study is worthy of investigation and having some environmental significance. However, the reviewer has provided additional comments for further improvement of the quality.

Comments:

Abstract: Include some quantitative data in the abstract, Also, at the end, add a statement on the important implications of this work.

Several studies have reported about the changes of abundance and dynamics of soil microbial communities with plant root exudates. A brief discussion is needed on this topic.

In the Introduction, the novelty and importance of this study with respect to existing knowledge in literature should be clearly highlighted.

At the end of discussion, add a section discussing the major implications and limitations of this work.

Delete all the pronouns like "we, ours, etc." and change the text accordingly.

The plant name and microbial species name should be written in italic text.

Table 1: In the footnote, explain all the abbreviations reported in the table.

Figure 4: In addition to this figure, is it possible to show the strength of correlation in a table containing the quantitative information.

There are several issues related to spacing errors throughout the manuscript, e.g. line 67 "diversity[3].", line 68: "small scale[1, 4].", etc.

To improve readers understanding and readability, the legends of all figures should be well written with explanation of the abbreviations representing the samples info.

Authors have carefully revised the manuscript according to the comments, and the quality of the manuscript has improved. Although the authors have done a great deal of work, there are still some issues to be resolved before publication. Thus, I suggest some minor revisions and expect the revised paper published in the AEM journal.

The line numbers in the “Response to Reviewer Comments” do not match the line numbers of the highlighted content in the Marked Up Manuscript or the Manuscript Text File. The following suggestions are given based on the Marked Up Manuscript.

Line130-145: Suggested change the formatting of entire paragraphs in Result 2.1 so that the entire text is typographically consistent.

Line181-184: The writing format of symbols needs to be unified. Please revise “broad-leaved forest>coniferous forest” to “broad-leaved forest > coniferous forest” and revise similar content accordingly.

Line191-192: Please revise “ADONIS R=0.436, p=0.05” to “ADONIS: R=0.436, p=0.05”, and suggest providing the result of ADONIS or ANOSIM test and the stress value of the PCoA analysis in Figure 3.

Line203-205: Please revise “N=82, E=318” to “N=82, E=318” and revise similar content accordingly.

Dear Editor, Dear reviewers

Thank you for your letter. I really appreciate all your comments and suggestions! These suggestions have enabled us to improve our work. Based on the instructions provided in your letter, we uploaded the file of the revised manuscript. Accordingly, we have uploaded a copy of the original manuscript with all the changes highlighted by using the track changes mode in MS Word. Appended to this letter is our point-by-point response to the comments raised by the reviewers. The comments are reproduced and our responses are given directly afterward in a different color (yellow). We would like also to thank you for allowing us to resubmit a revised copy of the manuscript. Thanks again!

Reviewer comments:

Reviewer #7 (Comments for the Author):

In this study, authors have investigated the impacts of pure forest and mixed forest types on the soil microbial dynamics. Overall, this study is worthy of investigation and having some environmental significance. However, the reviewer has provided additional comments for further improvement of the quality.

Comments:

Abstract: Include some quantitative data in the abstract, Also, at the end, add a statement on the important implications of this work.

Response:

Thanks for such important suggestion. As requested, the abstract has been revised with additional quantitative results added to it and the implication of this work supplemented.

Several studies have reported about the changes of abundance and dynamics of soil microbial communities with plant root exudates. A brief discussion is needed on this topic.

Response:

Thanks for such an insightful suggestion. The content has already been added in lines 92 - 95 of the text.

In the Introduction, the novelty and importance of this study with respect to existing knowledge in literature should be clearly highlighted.

Response:

We are thankful to the reviewer for this suggestion. This content has already been included in the text in lines 83 - 89.

At the end of discussion, add a section discussing the major implications and limitations of this work.

Response:

We thank the reviewer for the suggestion. This part has already been supplemented in lines 440-450 of the text.

Delete all the pronouns like "we, ours, etc." and change the text accordingly.

Response:

Thanks for mentioning this point. That type of inappropriate wording in the text has been corrected.

The plant name and microbial species name should be written in italic text.

Response:

We thank the reviewer for highlighting this point. All species names have been revised to Italics in the full text.

Table 1: In the footnote, explain all the abbreviations reported in the table.

Response:

Following your suggestion, we have revised footnotes and explained all the abbreviations reported in in the tables and figures.

Figure 4: In addition to this figure, is it possible to show the strength of correlation in a table containing the quantitative information.

Response:

Thanks for such important suggestion. However, since the amount of data is quite large, we processed it into the form of a heat map. A graph (Figure 5) has been added to clearly demonstrate the quantitative levels of correlation between the results.

There are several issues related to spacing errors throughout the manuscript, e.g. line 67 "diversity[3].", line 68: "small scale[1, 4].", etc.

Response:

We thank the reviewer for the indicating this crucial point. All spacing errors throughout the manuscript have been corrected.

To improve readers understanding and readability, the legends of all figures should be well written with explanation of the abbreviations representing the samples info.

Response:

In the light of your suggestion, we have added footnotes for all tables and figures of this manuscript.

Re: Spectrum02642-23R2 (Compared with pure forest, mixed forest alter microbial diversity and increase the complexity of inter-domain networks in arid areas)

Dear Dr. lu gong:

Your manuscript has been accepted, and I am forwarding it to the ASM production staff for publication. Your paper will first be checked to make sure all elements meet the technical requirements. ASM staff will contact you if anything needs to be revised before copyediting and production can begin. Otherwise, you will be notified when your proofs are ready to be viewed.

Sincerely,
Diyan Li
Editor
Microbiology Spectrum